

# An update of data compilation on the biological response to ocean acidification and overview of the OA-ICC data portal

Yan Yang[1, 2], Patrick Brockmann[3], Carolina Galdino[2], Uwe Schindler[4], Frédéric Gazeau[5]

[1]State Key Laboratory of Marine Environmental Science and College of Ocean and Earth Sciences, Xiamen University,
Xiamen, 361102, China
[2]International Atomic Energy Agency, Environment Laboratories, 4a Quai Antoine 1er, 98000, Monaco
[3]Laboratoire des Sciences du Climat et de l'Environnement / Institut Pierre Simon Laplace, LSCE/IPSL, CEA-CNRS-UVSQ, Université Paris-Saclay, Gif-sur-Yvette, 91191, France
[4]MARUM, Center for Marine Environmental Sciences, University of Bremen, Bremen, 28359, Germany
[5]Sorbonne Université, CNRS, Laboratoire d'Océanographie de Villefranche, LOV, Villefranche-sur-Mer, 06230, France

*Correspondence to*: Yan Yang (yangyan@xmu.edu.cn)

**Abstract.** Studies investigating the effects of ocean acidification on marine organisms and communities are increasing every year. Results are not easily comparable since the carbonate chemistry and ancillary data are not always reported in similar
units and scales, and calculated using similar sets of constants. To facilitate data comparison, a data compilation hosted at the data publisher PANGAEA was initiated in 2008 and is updated on a regular basis (https://doi.pangaea.de/10.1594/PANGAEA.962556, Ocean Acidification International Coordination Centre, 2023). By November 2023, a total of 1501 data sets (over 25 million data points) from 1554 papers have been archived. To easily filter and access relevant biological response data from this compilation, a user-friendly portal was launched (https://oa-icc.ipsl.fr)
in 2018. Here we present the updates of this data compilation since its second description by Yang et al. (2016) and provide an overview of the "OA-ICC portal for ocean acidification biological response data" launched in 2018. Most of the study sites from which data have been archived are in the North Atlantic Ocean, North Pacific Ocean, South Pacific Ocean and Mediterranean Sea, while polar oceans are still relatively poorly represented. Mollusca and Cnidaria are still the best represented taxonomic groups. The biological processes most reported in the datasets were growth and morphology. Other
variables that can potentially be affected by ocean acidification and are often reported include calcification/dissolution, primary production/photosynthesis, and biomass/abundance. The majority of the compiled datasets have considered ocean acidification as a single stressor, but their relative contribution decreased from 68% before 2015 to 57% today, showing a clear tendency towards more data archived from multifactorial studies.

## 1 Introduction

Ocean acidification refers to the rapid change of seawater chemistry, such as an increase in the partial pressure of carbon dioxide ($p$CO$_2$) and dissolved inorganic carbon as well as a decrease in pH and in the saturation state of seawater with



respect to calcium carbonate, arising from the uptake of excess anthropogenic $CO_2$ by the ocean (Orr et al., 2005; Feely et al., 2004; Gattuso et al., 2014). These changes will impact marine organisms and communities, and will alter marine ecosystems in ways that are still under active investigation (Kroeker et al., 2013; Doney et al., 2020).

The number of papers addressing biological responses to ocean acidification has grown exponentially from 2004 to 2022 (25 and 465 papers per year, respectively) (Gattuso and Hansson, 2011; Ocean Acidification International Coordination Centre, OAI-ICC bibliographic database, http://www.tinyurl.com/oaicc-biblio). However, results are not always easily comparable since data are either not publicly available or archived in different data repositories in varying formats. Furthermore, carbonate chemistry and ancillary data are not always reported in similar units and scales, and calculated using similar sets of

constants. In response to this problem, a data compilation hosted at PANGAEA Data Publisher for Earth & Environmental Science (Felden et al., 2023) was initiated by the European Network of Excellence for Ocean Ecosystems Analysis (EUR-OCEANS) and the European Project on Ocean Acidification (EPOCA) in 2008 (Nisumaa et al., 2010), and is maintained in the framework of the International Atomic Energy Agency (IAEA) project OA-ICC in collaboration with Xiamen University and the Laboratoire d'Océanographie de Villefranche, France, since 2013. The goal of this data compilation is to ensure the

archival and streamlining of data on the biological response to ocean acidification (and other environmental drivers) from published articles, as well as to provide easy access to the data for all users. To easily filter and access relevant biological response data from this compilation, a user-friendly portal was launched (https://oa-icc.ipsl.fr) in 2018. Between April 2012 and November 2023, datasets in the OA-ICC data compilation were viewed by users for 12684 times and downloaded for 5466 times. We report here on the updates of this data compilation since its second description by Yang et al. (2016) and

provide an overview of the "OA-ICC portal for ocean acidification biological response data" that was created a few years ago (https://oa-icc.ipsl.fr/).

## 2 Compilation process

The compilation process described in Nisumaa et al. (2010) and Yang et al. (2016) was followed to maintain consistency. Briefly, published papers focused on the biological response to ocean acidification are identified by searching the OA-ICC

news stream (http://news-oceanacidification-icc.org/) or through the OA-ICC bibliographic database for older papers. Data are extracted directly from tables or figures in these papers, or downloaded from other data repositories such as the Biological and Chemical Oceanography Data Management Office (BCO-DMO, https://www.bco-dmo.org/), the British Oceanographic Data Centre (BODC, https://www.bodc.ac.uk/) and the Australian Antarctic Data Centre (AADC, https://data.aad.gov.au/). Data not readily available in the papers or other data repositories are requested from the authors by

email. Data are not archived without the approval of the authors. Authors are also asked to fill in a quick survey (https://goo.gl/forms/qoBzcBkApTF9UJus2) in order to help the data curator to choose the proper keywords used in the data portal (see section 3). Data from papers that report less than two carbonate chemistry parameters are not included in the compilation. As part of the effort, the carbonate system variables are recalculated in a consistent way (see Nisumaa et al.





2010). Authors are always contacted to quality-check their data sets before archiving. Each data set has a citable DOI and is
publicly available on PANGAEA.

## 3 Data portal description

In order to facilitate data selection in the database and therefore to allow users to filter the compiled information based on
their research needs, there was a strong need to improve the way datasets are categorized. Building on community input that
was received during the 2016 Ocean in a High-CO$_2$ World Symposium in Hobart, Australia and the SOLAS-IMBeR Ocean
Acidification Working Group (https://imber.info/science/regional-programmes-working-groups/ocean-acidification-sioa/), a
set of keywords was established in collaboration with experts in the field and added to the data sets included in the
compilation. The keywords were designed to ensure that users are able to search and extract the datasets of interest,
ultimately facilitating data comparison and synthesis. The list of categories and their associated keywords are shown in Fig.
1.


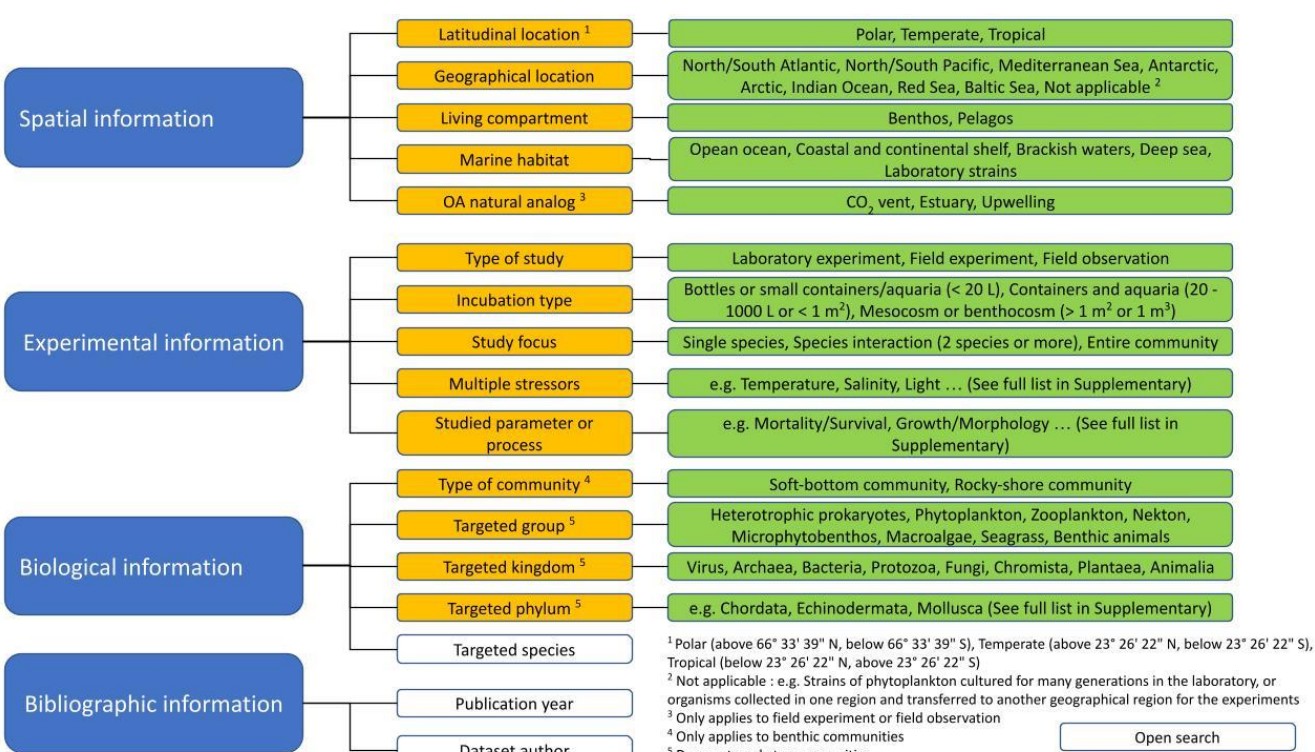

**Figure 1: List of categories and their associated keywords added to the data sets included in the OA-ICC compilation.**



The data portal was launched in 2018 and has been updated on a monthly basis since then. The data portal allows users to filter data sets according to their research interests and provides a page presenting the "User instructions". Briefly, users have the possibility to filter datasets based on keywords grouped in three categories: spatial information, experimental information and/or biological information. In the "Filter datasets" tab, users have access to bar plots showing the number of datasets tagged with each keyword and under each category. All keywords can be clicked in order to filter the datasets. Furthermore, users can directly select the species of interest from the keyword "Targeted species", by clicking on their names in the popup menu or entering the species name(s) in the search window. Finally, users can also filter the datasets by publication year and/or dataset author by selecting the years or author names from the popup menu or enter year/author name in the search window.

The number of selected results displays close to the "Selection" tab. Once selection has been finalized, matching datasets can be downloaded on this page as a single compressed file on the user local disk. The included data files can be opened using a text editor or any spreadsheet program such as Excel or OpenOffice.

A list of papers included or not included in this database is provided in the "Included/not included papers" tab, showing the full citations of these papers including a clickable doi (when available) and the doi of the corresponding datasets on PANGAEA (when archived). The process to generate this list is as follows: 1) a file shared with the OA-ICC bibliographic database team indicates if data from new papers were archived or not, 2) corresponding doi and keywords are added to the bibliographic database (OAICCdb for papers from which data were archived - only for these papers a unique doi/PANGAEA number is allocated; OAICCnoanswer when data could not be retrieved; OAICCincomplete when less than two parameters of the carbonate chemistry are provided and OAICCdatalost when data were lost), 3) an extraction of the bibliographic dataset following the selection of these keywords and an update of the publication list on the OA-ICC data portal is performed on a quarterly basis.

## 4 Data summary

Until now (2008-2023), 3665 relevant papers have been identified. A total of 1501 datasets (over 25 million data points) were archived from 1554 of these papers. Data from companion papers can be combined into a single data set (e.g. https://doi.pangaea.de/10.1594/PANGAEA.942466, Gazeau et al., 2021), which explains the discrepancy between the number of papers and data sets. Data of the remaining 2111 papers could not be added to the compilation for the following reasons: (1) less than two carbonate system parameters were measured in 573 papers, preventing the calculation of the carbonate chemistry; (2) data from 1522 papers could not be obtained from the authors; and (3) data from 16 papers were lost by authors (Fig. 2).

The earliest data included in the compilation were published in 1967 (Traganza, 1967). No data from the papers published during the period 1968-1993 could be archived because data were lost or could not be obtained from the authors. Data from 41 papers published between 1994 and 2006 were archived. The quantity of data archived for a given year follows the



increase in publication rate of biological response papers, with data from 13 papers archived for 2007 to 147 papers for 2013. On average during the period 2014-2022, data from 118 papers were archived each year. Data from 1015 papers (68% of the total number included in the compilation) have been archived since the last update presented in Yang et al. (2016).

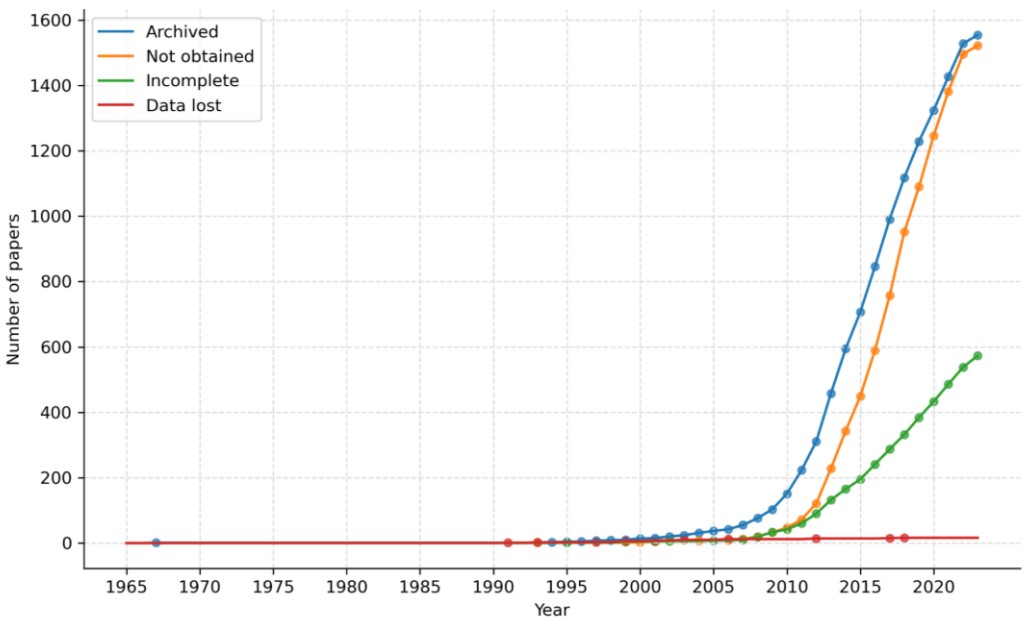


**Figure 2: Cumulative number of papers for which data have been included in the compilation ("Archived"), papers for which data could not be obtained ("Not obtained"), papers which reported less than two carbonate system parameters ("Incomplete") and papers for which the data have been lost ("Data lost"). The x-axis corresponds to the publication year.**

In order to produce the figures presented in the following sections, keywords describing the geographical location, study focus, targeted phylum, studied parameter or process, and multiple stressors were extracted from datasets for further analyses. Information on the country/region of affiliation of the first author was retrieved from the companion OA-ICC bibliographic database. Finally, the pair of carbonate system variables that was used for computations in seacarb is identified by Carbonate System Computation (CSC) flags (Table 1 in Nisumaa et al., 2010). This flag was used to investigate the percentage of

datasets considering one or another pair of the carbonate system. All python notebooks that were used to create the different figures of this article are publicly available (https://zenodo.org/records/8366844, Brockmann, 2023).

### 4.1 Geographical coverage

In the OA-ICC data compilation, the location of study sites indicates where the studied organisms were collected or the location of the natural communities investigated. If the geographical region was not clearly indicated in the paper or

organism(s) considered no longer representative of the study area, e.g. strains of phytoplankton cultured for many

generations in the lab, or organisms collected in one region and transferred to another geographical region for the experiments, datasets are categorized as "Not applicable" which comprises 198 datasets (13% of the total number of datasets; Fig. 3). The best covered geographical areas are the North Atlantic Ocean, North Pacific Ocean, South Pacific Ocean and Mediterranean Sea (420, 329, 244, 127 datasets, respectively). The Baltic Sea, Arctic Ocean, Antarctic Ocean, Indian Ocean,

South Atlantic Ocean, and Red Sea collectively represent only 15% of the datasets. Although more data of studies performed in the Antarctic (34 datasets) and the Arctic (26 datasets) were archived since 2015, polar oceans are still relatively poorly represented in the data compilation considering their strong vulnerability to ocean acidification (Orr et al., 2005; Steinacher et al., 2009).

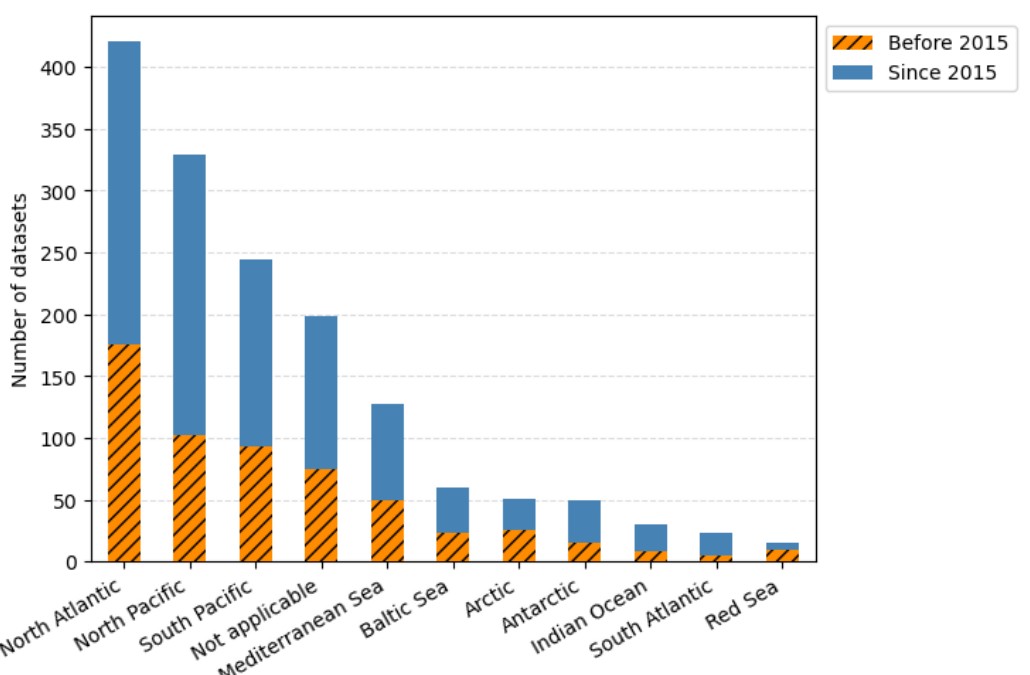


**Figure 3: Geographical coverage of datasets included in the OA-ICC data compilation, compared to those archived prior to 2015.**

## 4.2 Taxonomic coverage

Most of the studies from which datasets were archived investigated the impact of ocean acidification on single species (1218

datasets) while relatively few considered entire communities (262) and species interaction (59; Fig. 4a). Mollusca (330 datasets, 22% of total datasets) and Cnidaria (204 datasets, 14%) are the best represented taxonomic groups (Fig. 4b). A total of 117 datasets on Chordata were archived since 2015, making the relative number going up from 7% to 10%. Arthropoda (118), Ochrophyta (102) and Haptophyta (97) are also well represented, followed by Echinodermata (89), Rhodophyta (74)

and Chlorophyta (50). The amount of datasets concerning Foraminifera, Cyanobacteria, Myzozoa, Tracheophyta,, Annelida,
Porifera, Bryozoa, Charophyta, Proteobacteria, Brachiopoda, Platyhelminthes and Xenacoelomorpha are relatively low (1-32 datasets).

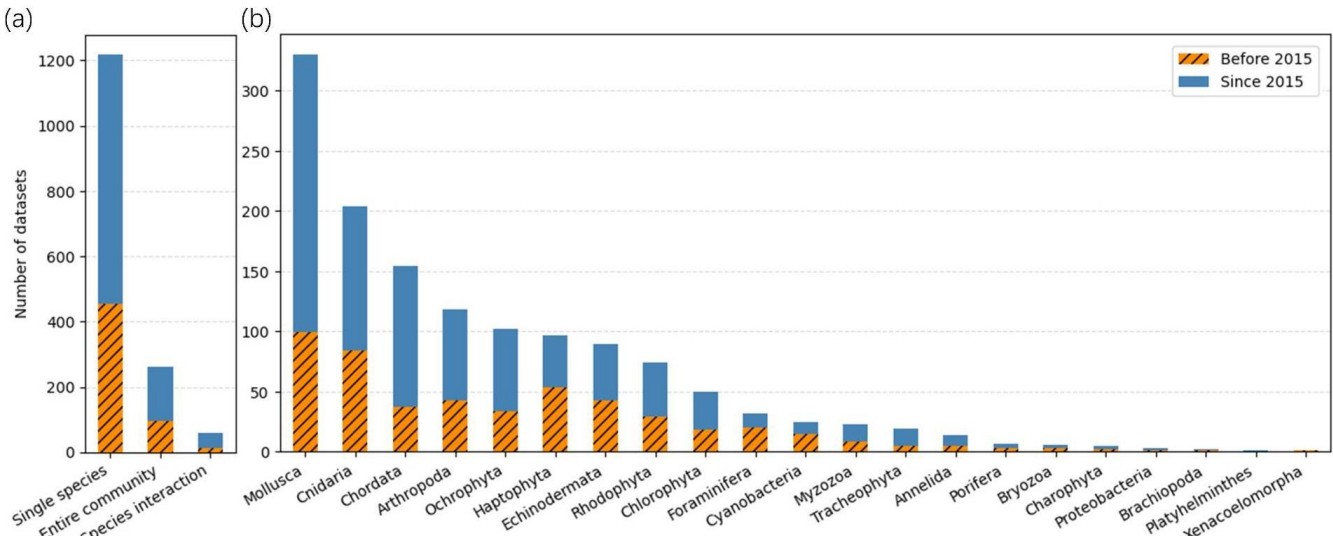

**Figure 4: (a) Study focus and (b) taxonomic coverage of datasets included in the OA-ICC data compilation, compared to those archived prior to 2015.**


### 4.3 Biological processes

The biological processes most reported in the datasets were growth and morphology (706 datasets, 47% of total datasets; Fig. 5). Other variables that can potentially be affected by ocean acidification and are often reported include calcification/dissolution (376, 25%), primary production/photosynthesis (361, 24%), biomass/abundance/elemental

composition (347, 23%), respiration (291, 19%), behaviour (214, 14%), mortality/survival (195, 13%), reproduction (175, 12%) and community composition and diversity (150, 10%). The data compilation also comprises datasets reporting on biological variables categorized as "other metabolic rates" and "other studied parameter or process" (159 and 194 datasets, respectively). The first category comprises processes such as nitrogen fixation, ammonia excretion, enzyme activities, while the second comprises variables/processes such as mechanical properties, bleaching, isotopic fractionation (see detailed

description of keywords in supplementary). Some datasets have also reported on variables such as gene expression (incl. proteomics), acid-base regulation, development and immunology/self-protection (26-91 datasets). Calcification/dissolution are less represented today (25%) than it was before 2015 (33%), indicating the initial imbalance (of considerably more datasets on calcification than on other processes) has further improved.



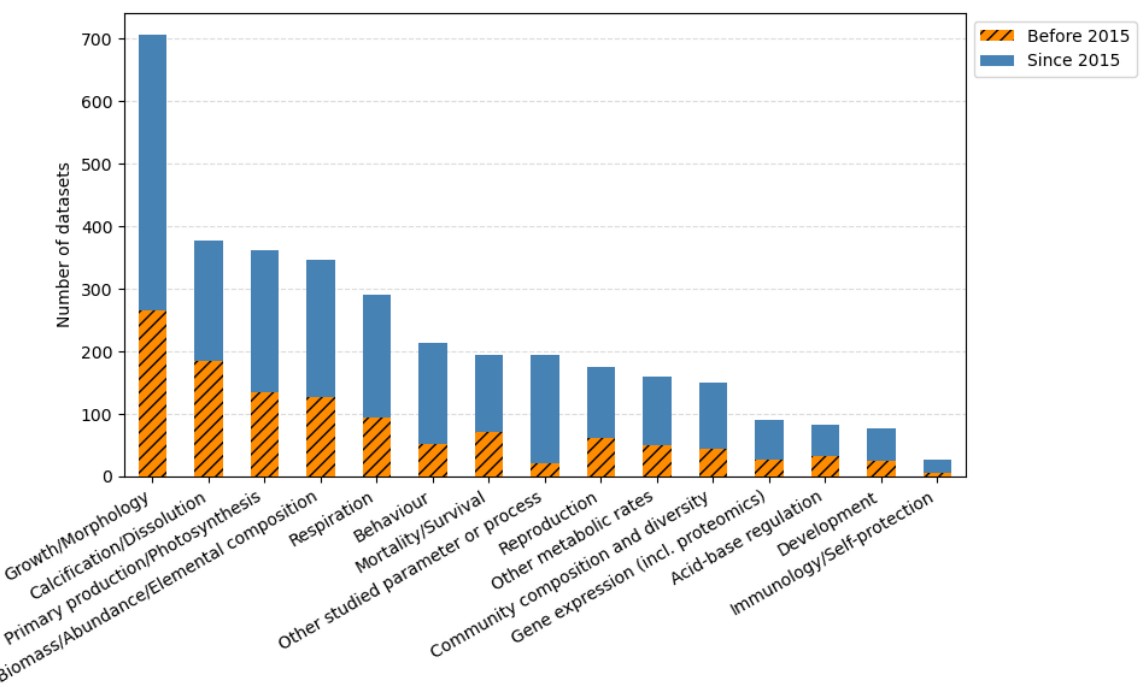

**Figure 5: Biological processes reported in the datasets included in the OA-ICC data compilation, compared to those archived prior to 2015.**

## 4.4 Multiple factors

The majority of the compiled datasets have considered ocean acidification as a single stressor, but their relative contribution decreased from 68% before 2015 (Yang et al., 2016) to 57% today, showing a clear tendency towards more data archived from multifactorial studies. The main other factors studied in addition to ocean acidification are temperature (370 datasets, 25%), light (99 datasets, 6%) and macro-nutrients (62 datasets, 4%; Fig. 6). Relatively few studies for which data were archived have reported on multi-stressor studies including oxygen, inorganic toxins, salinity, micro-nutrients and organic toxins (9-39 datasets). There are 111 datasets that have also reported on the combined effect of changes in carbonate chemistry with other factors such as food supply, water flow, calcium ion concentration, presence/absence of the predator cue, etc.




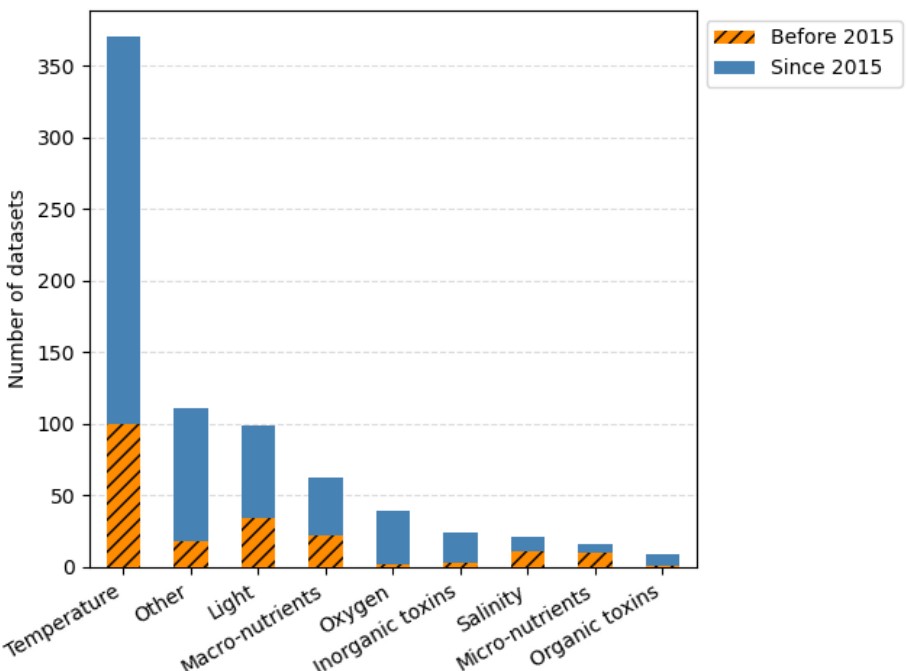

**Figure 6: Datasets of papers that have manipulated the carbonate chemistry as well as other variables.**

## 4.5 Countries/regions of first-author affiliation

Based on first-author affiliation, a total of 47 countries/regions contributed to the papers from which data were archived. The largest number of papers originates from European countries (717 papers, 46%; Fig. 7a). Within Europe, most of the papers were published by German and UK scientists (243 and 128 papers, respectively; Fig. 7b). USA (327 papers, 21%), China (165 papers, 11%) and Australia (147 papers, 9%) also significantly contribute to the data compilation. During 2011-2023, the total number of papers originating from South American countries increased from 1 (Panama) to 41 (Chile, Brazil, Mexico, Argentina, Colombia, Panama and Cuba). Data from papers published by scientists from Southeast Asia (Malaysia, Philippines) and Africa (South Africa and Angola) were added after the update in Yang et al. (2016).

There is still data from many papers which could not be obtained from the authors. From these missing papers, 82% are from European countries, USA, Australia and China (477, 341, 219 and 212 papers, respectively).

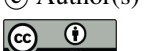



**Figure 7: Countries/regions of affiliation of first author of papers from which data were archived (a) all over the world and (b) in Europe.**


## 4.6 Measured carbonate chemistry variables

Total alkalinity ($A_T$) is the carbonate chemistry variable that is still the most measured (84% of the data sets; Fig. 8). The other variables measured include pH (76%), dissolved inorganic carbon ($C_T$, 34%) and the partial pressure of carbon dioxide ($p$CO$_2$, 7%). Out of the 76% datasets that measured pH, 39% reported pH on the total scale, 37% reported pH on National




Bureau of Standard (NBS) scale, seawater scale (SWS) or free scale. There is a clear tendency towards more datasets with pH reported on the total scale since 2015, since the ratio of total scale to other scale increased from 0.75 pre-2015 (Yang et al., 2016) to 1.0 today. The pH value on the total scale is generally lower and higher than on the NBS/NIST scale and the SWS scale, respectively (Dickson, 2010), which makes the direct comparison of experimental results difficult. In our compilation, all other scales are converted to the total scale as recommended in the Guide to Best Practices in Ocean

Acidification Research and Data Reporting (Dickson, 2010).

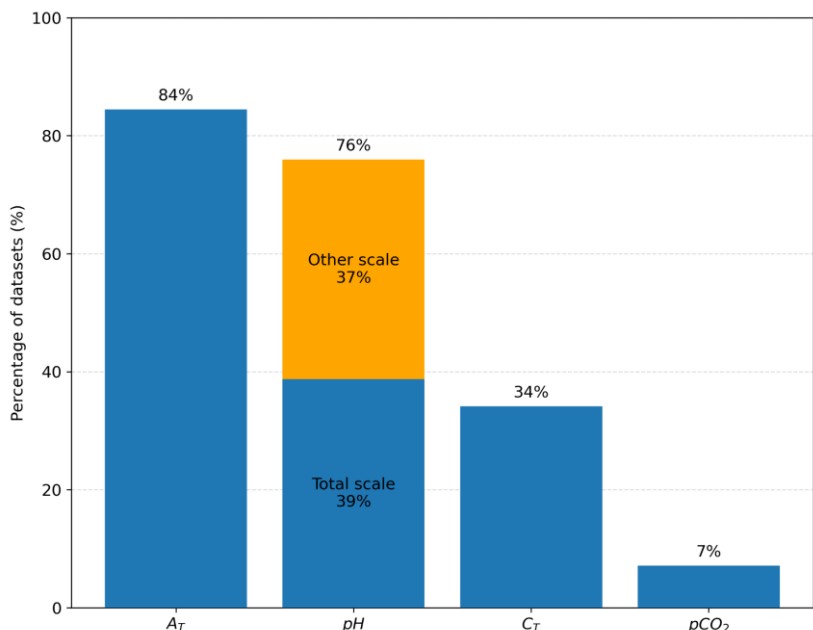

**Figure 8: Variables of the seawater carbonate system reported in the data sets.**

## 5 Conclusions and future directions

The compilation of data related to the biological response of marine organisms and communities to ocean acidification (and other drivers) launched in 2008 and pursued during the EPOCA European project, has continued uninterrupted since then thanks to direct and in-kind contributions from several IAEA Member States. The amount of data archived in our database has grown substantially in the last years, following the important increase of studies conducted in this field of research. However, while until 2015, 53% of the relevant papers could be included in the compilation, this proportion has dropped

significantly in the last few years, reaching 42% until 2023. As mentioned previously, our strategy, when relevant data are not directly available in the published manuscripts (or in the supplementary material), is to request datasets from corresponding authors. Although data curation is a fairly fast process (datasets are requested to the authors a few weeks to a



few months after publication), it is unfortunately common to have troubles contacting corresponding authors who have
changed institutions. This is, most of the time, not unsolvable as finding the email addresses of co-authors is possible but add
to the time lapses and the efforts necessary to archive data, as receiving data usually requires sending one or more reminders
and several email exchanges with the author to obtain all relevant data. Adding the contact email of a permanent researcher
in the published manuscript could definitely help optimizing the process. Our response rate, although low, is not far from
what is typically found in any research field (28-56%; Tedersoo et al., 2021). On very minor occasions, authors declined
sharing their data due to ethical/legal issues or a concern of misuse. For the large majority of the time datasets cannot be
collected, the reason is a lack of answer from the authors after three reminders. A growing number of journals now request
publishing datasets upon acceptance of manuscripts, let us hope this will increase data availability in the coming years.

The compiled database highlights the increase in studies not only focused on ocean acidification but also including other
relevant environmental drivers. Since, to the best of our knowledge, no efforts at the international level have been done on
compiling all relevant data on the marine biological response to some of the most studied environmental drivers, ocean
warming and deoxygenation (e.g. Sampaio et al., 2021), a desirable evolution of our database in the coming years would be
to include these studies and corresponding datasets in the same way we have been collecting data on ocean acidification
since 2008.

Since the last description of the OA-ICC database (Yang et al., 2016), the geographical coverage of compiled datasets did
not significantly change with still a clear under-representation of the southern hemisphere and polar regions. The last
international symposium on the Ocean in a high $CO_2$ world organized in Lima (Peru) in 2022 demonstrated a growing
activity of southern countries in this field of research, especially from South America, that will undoubtedly help fill this gap
in the coming years. Conducting research in polar regions is obviously more difficult in such harsh and remote environments
and requires strong international cooperation, extended planning horizons, sizable budgets and long-term investment
(Figuerola et al., 2021). However, as the rate at which seawater is acidifying and warming is much higher in these regions
than anywhere in the global ocean, there is a strong need to substantially increase our research efforts in these threatened
environments.

**Data availability**

Access to data presented in this paper is at https://doi.pangaea.de/10.1594/PANGAEA.962556 (Ocean Acidification
International Coordination Centre, 2023). The OA-ICC database exposed from the OA-ICC web portal described previously
is originally available from the PANGAEA portal (https://www.pangaea.de/?q=Project:OA-ICC) and accessible from
Elasticsearch requests such as http://ws.pangaea.de/es/dataportal-oa-icc/pansimple/_search?size=20 for a simple request to
fetch the 20 first datasets metadata. Metadata harvesting can also be achieved from the use of the Open Archives Initiative
Protocol          for          Metadata          Harvesting          (OAI-PMH)          with          request          like

http://ws.pangaea.de/oai/provider?verb=GetRecord&metadataPrefix=pan_md&identifier=oai:pangaea.de:doi:10.1594/PANGAEA.903033

**Code availability**

All python notebooks that gather the web scraping, metadata harvesting and the processing codes used to create the different figures of this article are publically available. Those notebooks are embedded in a jupyter binder container that allows anyone to run and inspect the different scripts. They are published on the Zenodo repository with the following DOI:

https://zenodo.org/records/8366844 (Brockmann, 2023).

**Author contributions**

FG is the OA-ICC focal point for data management. YY is the data curator for the OA-ICC data compilation. PB maintained and updated the OA-ICC data portal. CG maintained the OA-ICC bibliographic database. US provided the usage statistics of this data compilation. All authors contributed to the manuscript.

**Acknowledgements**

This effort was initiated by Jean-Pierre Gattuso, Laboratoire d'océanographie de Villefranche (CNRS/SU), France, as part of the EU projects EUR-OCEANS and EPOCA. We would like to thank previous or current contributors to the data compilation, Olga Anghelici, Trevor Eakes, Frank Graba, Lina Hansson, Marine Lebrec (funding through OA-ICC), Anne-Marin Nisumaa (funding through EPOCA and OA-ICC) and Nassim Taalba (funding through EUR-OCEANS). Thank you

to the members of the SOLAS-IMBeR Ocean Acidification Working Group (SIOA) for fruitful discussions. We are grateful to all the authors for the provision of data and to the staff of PANGAEA for the considerable help and support.

**Financial support**

This research work has been funded by the US through the Peaceful Uses Initiatives (PUI) program under the project "Ocean Acidification International Coordination Centre (OA-ICC), Phase III", implemented within the framework of the IAEA

Department of Nuclear Sciences and Applications. The IAEA is grateful to the Government of the Principality of Monaco for the support provided to the Marine Environment Laboratories. This effort has also been supported by the National Natural Science Foundation of China Basic Science Center Program (grant no. 42188102).



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
