# Peer review of "An update of data compilation on the biological response to ocean acidification and overview of the OA-ICC data portal"

_Earth System Science Data, 2023_

## Author Response (AR1)

**Reply on referee #1**

**Overview paragraph/general comments**:

Overall this is a good paper that describes the data collection and curation of this useful biological dataset. Moving forward the research community is going from individual experiments to meta analyses and data sets/compilations like the one described in this paper are very important to streamline/standardize those efforts. The data quality is good, methods used follow best practices and are plausible with no detectable faults. The data presentation is good and the authors use freely available python code to interrogate the data. Below are a few minor revisions for the authors to consider.

[**Response**] We appreciate the positive comments from the referee. The detailed revisions are described below responding to the comments and suggestions from the referee.

**Individual scientific questions/issues/specific comments:**

The definition of OA given in the first sentence of "rapid change" is qualitative and while there are citations at the end of the sentence none of those citations specifically state what "rapid change" means. I suggest either clarifying that the author is referring to rapid on the geologic scale or striking the word rapid in the first sentence all together.

[**Response**] Accepted. We have deleted "rapid" in the sentence as "Ocean acidification refers to the change of seawater chemistry, …" in Lines 30 of the revised MS.

https://www.nature.com/articles/s41467-024-47064-3.pdf I realize that this article just came out however I think it is relevant and suggest the authors consider including this reference in their discussion. Perhaps in line 235.

[**Response**] Accepted. We have added the citation of the suggested reference in Line 234 of the revised MS as "...no efforts at the international level have been done on compiling all relevant data on the marine biological response to some of the most studied environmental drivers, ocean warming and deoxygenation (e.g. Alter et al., 2024; Sampaio et al., 2021),..."

**Technical comments:**

Line 218 "has grown substantially in the last years..." Suggest adding in the number of years you are referencing to make this statement more specific

[**Response**] Accepted. We have specified the number of years in the statement as "The amount of data archived in our database has grown substantially in the last 15 years, in keeping with the increase of studies conducted in this field of research." in Line 218 of the revised MS.

Same with line 220 "significantly in the last few years" can you state explicitly what you mean by few? It appears from the text you mean between 2015 and 2023. Please clarify this in the text.

[**Response**] Accepted. We have modified the statement as "However, while until 2015, 53% of the

relevant papers could be included in the compilation, this proportion has dropped significantly in the last 8 years, reaching 42% until 2023." in Line 220 of the revised MS.

Line 222 "(datasets are requested to the authors...) suggest revising to read (dataset requests are sent to the authors....)

[**Response**] Accepted. This sentence has been rephrased as "Although data curation is a fairly fast process (dataset requests are sent to the authors a few weeks to a few months after publication), ..." in Line 222 of the revised MS.

Line 224-225 "...co-authors is possible but add to the time lapses and the efforts necessary to archive data..." suggest revising to "...co-authors is possible but significantly adds to the time and effort required to archive data,..."

[**Response**] Accepted. This sentence has been rephrased as "This is, most of the time, not unsolvable as finding the email addresses of co-authors is possible but significantly adds to the time and effort required to archive data,..." in Line 224-225 of the revised MS.

Line 226 "...obtain all relevant data." Suggest adding and metadata to the end of this sentence to read "...obtain all relevant data and metadata."

[**Response**] Accepted. We have added "metadata" to the end of the sentence as "... as receiving data usually requires sending one or more reminders and several email exchanges with the author to obtain all relevant data and metadata." in Line 226 of the revised MS.

Line 227 should be optimize not optimizing

[**Response**] Accepted. We have revised the sentence as "Adding the contact email of a permanent researcher in the published manuscript could definitely help optimize the process." in Line 227 of the revised MS.

**Reply on referee #2**

This manuscript provides a complete and timely update on the ocean acidification data base hosted by PANGEA, along with an overview of the OA-ICC portal for ocean acidification biological response data that was launched in 2018. The motivation for the study is clear and the necessary background to understand the project is given in the introduction. The results provide a clear and meaningful overview of the datasets that have been compiled, their geographical distributions, taxonomic affinities, and other relevant groupings. Overall, a thorough and useful update of the OA database that will be of broad interest to the ocean acidification research community.

[**Response**] We appreciate the positive comments from the referee. The detailed revisions are described below responding to the comments and suggestions from the referee.

I suggest some minor edits that may assist with clarity and ease of interpretation.

Line 20: Change "the updates" to "an update"

[**Response**] Accepted. We have revised the sentence as "Here we present an update of this data compilation..." in Line 20 of the revised MS.

Line 42: This is awkwardly written. Maybe change "is maintained in the framework of" to "is maintained within the"

[**Response**] Accepted. We have revised the sentence as "In response to this problem, a data compilation…, and is maintained within the framework of the International Atomic Energy Agency (IAEA) project OA-ICC…" in Line 43 of the revised MS.

Line 48: Delete "for" before 12684 and before 5466

[**Response**] Accepted. We have deleted "for" in the sentence as "…, datasets in the OA-ICC data compilation were viewed by users 12684 times and downloaded 5466 times." in Line 49 of the revised MS.

Line 89: Should this be "local disk" or "local drive"

[**Response**] We have changed "local disk" to "local drive" which is a bit broader and can encompass any internal storage device in Line 89 of the revised MS.

Line 92: There are a lot of "includings" in this sentence. Maybe change "papers including a" to "papers as well as a"

[**Response**] Accepted. This sentence has been rephrased as "A list of papers…, showing the full citations of these papers as well as clickable DOIs (when available) and the DOIs of the corresponding datasets on PANGAEA (when archived)." in Line 92 of the revised MS.

Line 147: Reword "making the relative number go up from" to "increasing the relative number from"

[**Response**] Accepted. This sentence has been rephrased as "…, increasing the relative number from 7% to 10%." in Line 147 of the revised MS.

Line 149: Change "amount" to "number"

[**Response**] Accepted. We have revised the sentence as "The number of datasets concerning…is relatively low (1-32 datasets)." in Line 149-150 of the revised MS.

Line 207: For clarity, it would be better to say "The pH on the total scale is generally lower than on the NBS/NIST scale and higher than on the SWS scale."

[**Response**] Accepted. We have modified the statement as "The pH on the total scale is generally lower than on the NBS/NIST scale and higher than on the SWS scale (Dickson, 2010), …" in Line 207 of the revised MS.

Line 216: Insert a comma after 2008.

[**Response**] Accepted. A comma has been inserted after 2008 in Line 216 of the revised MS.

Line 218: Change "following the important increase" to "in keeping with the increase"

[**Response**] Accepted. We have revised the sentence as "The amount of data archived in our database…, in keeping with the increase of studies conducted in this field of research." in Line 218 of the revised MS.

Line 229: Change "time datasets cannot" to "datasets that cannot"
[**Response**] Thanks for the suggestion. To make it clearer, we prefer to rephrase this sentence as "For the majority of the time, datasets cannot be collected due to a lack of response from the authors after three reminders." in Line 229-230 of the revised MS.

Line 213: Change "let us hope" to "and we hope"
[**Response**] We think this comment is for line 231. Accepted. We have changed "let us hope" to "and we hope" in Line 231 of the revised MS.